# The Construction of an Evaluation System of Corporate Social Responsibility to Employees: An Empirical Study in the Chinese Clothing Industry

**Yanting Jing [1], Wei Zhang [2], Yongjun Tang [2,*] and Yuzi Zhang [3]**

[1]  School of Business, Jinling Institute of Technology, 99 Hongjing Avenue, Nanjing 211169, China
[2]  Business School, Hohai University, 8 Focheng West Road, Nanjing 211100, China
[3]  School of Public Administration, Hohai University, 8 Focheng West Road, Nanjing 211100, China
[*]  Correspondence: yjtang@hhu.edu.cn

**Abstract:** Recently, with the intensification of employee suicides in well-known international companies such as Facebook and Pinduoduo, people are paying more and more attention to the violation of employee rights and interests. As an important embodiment of safeguarding the legitimate rights and interests of employees, the corporate social responsibility to employees has become one of the focuses of academic discussions. The aim of this article is to build a corporate social responsibility evaluation system for employees for Chinese clothing companies. As a representative of labor-intensive enterprises, enterprises in the cloth industry often need to rely on the strength of their employees to create value more than ordinary enterprises. Therefore, it is of practical significance to study the corporate social responsibility of employees in the cloth industry. In addition, China is an important exporting country of clothing in the world, and its market environment is different from that of developed countries. Research with Chinese enterprises as samples may lead to different conclusions. Finally, unlike general CSR, the evaluation of employee CSR needs to consider the importance of subjective and objective factors. At this time, the use of the catastrophe progression method can more accurately evaluate the weight of each factor. The result of our research on 100 Chinese clothing companies shows that enterprises with higher rankings in clothing industry will fulfill social responsibility to employees better. The use of the catastrophe progression method to evaluate corporate social responsibility to employees can reduce errors caused by subjective steps such as assigning weights in Analytic Hierarchy Process (AHP) and improve the accuracy of evaluation.

**Keywords:** corporate social responsibility to employee; evaluation system; catastrophe progression method

## 1. Introduction

With the development of society and economy, investors around the world no longer only pay attention to the corporate benefits obtained by enterprises, but increasingly take corporate social responsibility (CSR) as an important indicator for evaluating enterprises. Valeri pointed out that management should guide the company in activities aimed at creating value not only for shareholders, but also for employees, suppliers and customers [1].

Among them, employee rights and interests are one of the most important elements to be considered in corporate social responsibility [2]. However, there are many misunderstandings in the current practice of corporate social responsibility in China. Some companies limit their social responsibility to donations and charitable activities for the sake of reputation or public and government relations and are reluctant to spend money on the improvement of employee rights and interests.

Especially under the current general trend of slow economic growth under the influence of COVID-19, some companies have cut costs by laying off a large number of

employees and violating the interests of employees. Even extreme incidents such as "Facebook employees jumped off the building [3]" and "Alibaba employees being abused [4]" occurred from time to time, causing many negative social impacts.

Therefore, this article will discuss the construction of the corporate social responsibility system at the employee level and try to strengthen the company's emphasis on corporate social responsibility to employees through system norms and promote the protection of employees' rights and interests.

In the field of corporate social responsibility, a relatively mature theoretical system has been formed. The theory of corporate social responsibility believes that enterprises should not only create profits for shareholders, but also undertake social responsibilities to creditors, employees, suppliers, customers, customers, communities and the natural environment, mainly including compliance with business ethics, safe production, occupational health and labor protection. These include activities such as protecting the legitimate rights and interests of people, protecting the environment, supporting charity and public welfare and protecting vulnerable social groups [5].

Additionally, the International Social Responsibility Guidelines Standard (ISO26000) also proposes that corporate social responsibility includes organizational management, human rights, labor, environment, fair management, consumer rights protection, community participation, social development and collaboration between stakeholders [6].

However, in terms of corporate social responsibility to employees, not only is it not valued in corporate practice, but the current international theoretical research is still relatively weak, which reflects that the importance of corporate social responsibility to employees needs to be improved.

Based on this, this article conducts an empirical study on the top 100 enterprises in China's clothing industry in 2018 through the catastrophe progression method. Through 61 quantitative indicators related to the corporate social responsibility to employees, the CSR performance of the top 100 enterprises is scored. Finally, we proved that companies with higher overall rankings, or financial performance, generally have higher levels of corporate social responsibility to employees. We believe that this may be related to the degree of development and the size of the company.

The contributions of this article are described in the following. First of all, on the research topic, we choose the perspective of corporate social responsibility to employees as the starting point of the research. At present, most scholars' research on corporate social responsibility focuses on the environment, taxation and integrity, but ignores employees' rights and interests. Secondly, in the selection of samples, we limit the scope of the sample to Chinese enterprises. Due to the different forms of production organization, employees in China, especially in private enterprises, are more likely to defend their rights and interests through self-struggle, and even some workers have no such awareness at all [7], which is inconsistent with the current status of industrial relations in most developed countries. Venturelli also pointed out that investors in China's financial market do not pay enough attention to corporate social responsibility, and this is mostly because of China's special market conditions. Chinese enterprises are often driven by external forces such as the government and regulatory authorities to fulfill their social responsibilities, rather than taking the initiative to undertake them from the perspective of enterprise development [8]. Studies of emerging countries may come to different conclusions due to different stages of development. Thirdly, we take China's clothing industry as the research object. The clothing industry is a labor-intensive industry, and the role of employees is more critical than other industries. In addition, China is a major clothing exporter in the world [9], and it is more representative to take Chinese enterprises as the research object. Finally, in the research method, we use the catastrophe progression method to determine the weight of the enterprise's evaluation index of employee social responsibility. Different from CSR evaluation from other perspectives, CSR evaluation at the employee level needs to be weighted according to the importance of the influencing factors, combined with

subjective and objective factors. The use of the catastrophe progression method can make the evaluation model proposed in this article more accurate.

The rest of the article is structured as follows. The second part reviews the existing research on the development of CSR theory, the CSR evaluation system and corporate social responsibility to employees and puts forward six basic viewpoints as the theoretical basis for the selection of some factors of the methodology. The third section describes the source of the data and the choice of methodology and its rationale. The fourth section introduces the application of the evaluation system and the evaluation results of 100 Chinese clothing enterprises. The fifth section discusses the findings and summarizes the link between firms' CSR performance and financial performance. The last section concludes the article and reveals some limitations of our study, making recommendations for future research.

## 2. Literature Reviews and the Construction Evaluation System of CSR

### 2.1. The Development of Corporate Social Responsibility

The concept of "social responsibility" was put forward by British scholar Oliver Sheldon in 1923 [10]. He connected corporate social responsibility with the responsibility of managers to meet all sorts of human needs both inside and outside the industry. He also believed that corporate social responsibility contained moral factors. Since then, the famous debate has been triggered between Dodd and Berle about whether the enterprise should bear social responsibility [11]. In the 1930s, with the development of enterprise theory represented by Coase [12], the western theoretical and practice circles began to critique "shareholders first", and thought the enterprise needs to bear certain social responsibilities at the same time of creating profits for shareholders, such as donations, improving employee welfare and taking some community service projects to give back to society. The original definition of social responsibility was put forward by Bowen in the book "*Social Business of the Businessman*" [13]. He thought that social responsibility was that businessmen had an obligation to make policy, make decisions or take some action in accordance with the goals and values of society's expectations.

In the 1970s, the frequent social issues of American corporations greatly promoted the development of the social responsibility movement. For example, Firestone's faulty tires caused large casualties at the time. P&G's menstrual pads problem made a lot of people die from the poison. At the same time, the environmental movement played a large role in promoting the development of corporate social responsibility from an environmental perspective. By the 1980s, the emergence and development of stakeholder theory represented by Freeman and Blair further promoted the development of the social responsibility theory and practice. In the book "*Strategic Management: A Stakeholder Approach*", Freeman defined stakeholder as "an organization's stakeholders are people or groups which can affect the realization of organizational goals or affected by its goals" [14]. Canada scholar Clarkon studied CSR performance from the perspective of stakeholders and classified stakeholders to construct an impact model on CSR performance [15]. Stakeholder theory has been clear that the objects which social responsibility contained are not only shareholders, but also employees, suppliers, customers, community, government and even the inanimate nature environment, which expanded the research scope of social responsibility.

Since the late 1990s, the study of social responsibility has turned to create the social responsibility standards which have practical guiding significance, such as promulgation and promotion of the corporate social responsibility standard SA8000, international social responsibility guidelines standard (ISO26000) and other measures promoting social responsibility of the most respected company ranking, etc. Since 2003, corporate social responsibility has caused widespread concern in China. In recent years, China has taken many measures to promote the fulfillment of social responsibility. For example, Shenzhen Stock Exchange (SSE) officially published "Listed Company Social Responsibility Guidance" in September 2006, which standardize the behavior of listed companies in fulfilling corporate social responsibilities and disclosing corporate information. "Labor Contract

Law", which was enforced on 1 January 2008, also further promoted the fulfillment of corporate social responsibility to employees [16].

However, at present, most scholars evaluate CSR mainly from the perspective of stakeholders, and pay more attention to the impact of corporate social responsibility on financial performance and decision-making of consumers and investors [17–19]. The factors considered are also more related to the environment, public welfare, tax payment, fraud, etc. [20–22]. For example, Park et al. decomposed corporate social responsibility into three specific responsibilities of economy, society and environment, and emphasized the interaction between corporate responsibility and consumer responsibility [23]. Gomez and Chalmeta's research also pointed out that corporate social responsibility plays an important role in consumers' evaluation of brands and products, and it is also of great significance for enterprises to improve economic efficiency [24].

### 2.2. Evaluation System of CSR

In the field of CSR evaluation, many scholars have already conducted a lot of research. For example, Liu et al. used DEMATEL to construct a network relationship diagram of four impact dimensions and 13 employee care standards, thereby constructing an employee social responsibility evaluation system [25]. Fassin and Buelens constructed the sincerity/hypocrisy index to describe the drivers of firm behavior, the intentions of actors, and the strength of firm communication [26]. Fang and Wang used the AHP-GRA model to provide an objective and reliable CSR evaluation model for automobile companies [27]. In the research of Aparici, Kapelko and Monge, their index design took into account the concept of Pareto efficiency, which promoted the theoretical study of the CSR evaluation model [28]. Rahdari, mainly from the perspective of the environment, using corporate governance, corporate social responsibility and corporate financial performance ratings, worked to establish a triangular rating system for evaluating corporate performance [29]. Venturelli et al. mainly adopted the fuzzy expert system (FES) to evaluate CSR [8]. In the above research, different researchers faced different research problems, adopted different research methods according to the characteristics of the research objectives, and also adjusted according to the market environment and industry characteristics of the samples.

The clothing industry is a very important industry for China, but it is also an industry that easily violates the rights and interests of employees. However, there are few studies on the social responsibility evaluation of the clothing industry. Only scholars such as Baskaran et al. have constructed the CSR of the Indian garment industry. His research considered factors such as discrimination, abuse of human rights, child labor, long working hours, unfair competition and pollution [30].

### 2.3. Corporate Social Responsibility to Employees

Current research has shown that companies do not attach much importance to Occupational Health and Safety (OHS) behavior, and only pay attention to a small amount of information that requires transparency and disclosure [31]. However, in fact, whether the rights and interests of employees are guaranteed is related to the production and operation of each link of the enterprise, and the social responsibility of the enterprise to the employees has a very important impact on the long-term development of the enterprise. With the increasing importance of human capital and the continuous outbreak of major social incidents caused by the violation of employees' rights and interests by enterprises in recent years [32], enterprises urgently need to study the social responsibility of employees. Through the scientific and quantitative evaluation model, the performance of the company can be seen more intuitively, and it is also more conducive to the follow-up research on employee satisfaction and corporate financial performance.

The theoretical basis for supporting enterprises to undertake corporate social responsibility to employees is mainly: human capital theory and employee rights theory. To sum up, the three basic ideas represented by human capital theory are as follows:

(1)  among many factors affecting economic development, humans are most critical and the economic development mainly depends on the improvement of people's quality rather than the abundance of natural resources or the amount of capital [33];

(2)  human capital is formed by investment [34];

(3)  investment in education and training of employees can bring more rewarding returns than the returns from investment in materials [35].

The basic views of employees' rights and interests theory are:

(1)  employees as people hold unique intrinsic value, and should have universal and equal human rights [36];

(2)  employees in the workplace should have all kinds of work-related rights, including five points as follows: right to not be unreasonably fired, the right of due process and fair treatment, the right of freedom, right of privacy, security and health in the workplace and right to non-discrimination [37];

(3)  employees' rights can create additional economic value for employers and companies, and employees can engage in meaningful work to maintain their autonomy and have certain permission to make decisions about the problems they encounter at work [37].

## 3. Method

### 3.1. Research Design and Sample

From the above research we can see that corporate social responsibility to employees is the core content of corporate social responsibility. Mcguire et al. divide corporate social responsibility activities into eight categories: the liability of manufacturing products, the liability of marketing activities, the responsibility of staff's education and training, the responsibility of protecting the environment, the responsibility of providing good welfare and a good relationship among employees, providing equal employment opportunities, paying attention to the safety and health of the employees and the responsibility of participating in charitable activities [38]. Half of them are directly related to employee social responsibility. It also further confirms the core status of the corporate social responsibility to employees. Kai Chang believed that corporate social responsibility by its nature is a corporate legal responsibility to society and mainly implied the responsibility undertaken for the internal adjustment of labor relations and the implementation of labor rights in terms of the scope of social responsibility [39].

The concentric circle theory about employee social responsibility is proposed in the report "Business Enterprise Social Responsibility" released by the U.S. Economic Development Conference and the inner circle shows the most basic social responsibility of corporate, namely job functions; the middle circle shows the secondary level of social responsibility, namely employee relations; the outside circle stands for the highest level of social responsibility, namely the emerging and undefined requirements of the employee social responsibility. Social Responsibility Standard SA8000 contains eight kinds of indicators:

(1)  child laborers;

(2)  forced employment;

(3)  healthy and safety;

(4)  freedom and right of collective bargaining;

(5)  differential treatment;

(6)  punitive measure;

(7)  working hours;

(8)  reward.

International Social Responsibility Standard ISO26000 contains seven aspects of corporate social responsibility:

(1)  human rights risk state;

(2)  guarantee basic rights;

(3)  promote employment and employment relationship;

(4)  working conditions and employment security;

(5)　　maintaining social dialogue;

(6)　　estimating work safety and health;

(7)　　participating in the human resources development and the site training.

With the aid of the above two standards and related literature and combined with the actual condition of our country, we construct the employee social responsibility evaluation index system.

In order to measure the performance of corporate social responsibility as comprehensively as possible, the indicators which we design contain the objective indicators and also join the subjective indicators as a supplement. The social responsibility evaluation index system is divided into two first-class indicators, which are employees' rights and the protection of human rights. Each indicator is decomposed into four sub-indicators at most, and we keep decomposing step by step until they can be quantified. According to the relative importance, we sort each level's indexes, which means the front of the row is relatively important and the relatively minor row will be behind. Specific factors and their order are shown in Table 1.

Finally, we decomposed the two large first-level indicators into 61 quantifiable indicators for the construction of the evaluation system to be carried out below.

This article takes the top 100 enterprises in China's clothing industry (by product sales revenue) in 2018 as the research sample. The ranking data of the top 100 enterprises in the clothing industry (by product sales revenue) in 2018 is from the China clothing association network (http://www.cnga.org.cn/, accessed on 26 July 2019) and the other data are customized from the website of the China Academy of Commerce (https://www.fxbaogao.com, accessed on 20 March 2020).

### 3.2. Evaluation Method

The current CSR evaluation methods are mainly the Delphi method, principal component analysis, entropy value method, analytic hierarchy process (AHP), etc. [40–42]. The comparison of findings and limitations of different research methods is shown in Table 2.

This article chooses catastrophe progression as this method only has a whole grasp for the relative weights of the same level without determining the weights of each indicator, while having many of the same effects of the principal component analysis and analytic hierarchy process. It can be said that the catastrophe progression method not only absorbs the advantages of the analytic hierarchy process, the effect function method and the fuzzy evaluation method, but also normalizes the difference set, thereby unifying the units of different properties and different measurement indicators. In addition, the evaluation results obtained by the catastrophe progression method are more real and accurate and can better reflect the performance of corporate social responsibility.

The catastrophe progression method is a comprehensive evaluation method. Firstly, it decomposes or groups the complicated and conflicting evaluation target. Then, it uses the catastrophe theory combined with fuzzy mathematics to produce the mutational and fuzzy membership degree function. Next, it uses a normalized formula to calculate a parameter that is the total membership function. After that, it will sort and analyze the evaluation indexes. The concrete steps of this method are as follows.

### 3.2.1. Decomposition Evaluation Targets

Using the catastrophe progression method, we can assess corporate social responsibility to employees. First of all, it is necessary to perform multi-level decomposition of the evaluation target, and the decomposed indicators have a tree-like structure. In order to facilitate the measurement and calculation of the evaluation, we can stop the decomposition when the indicators of these decompositions can be quantified. Since there are no more than 4 control variables in a general variation system, the bottom of the corresponding control variables should not exceed 4, so the catastrophe series method is more applicable and accurate when dealing with multi-criteria decision-making problems.

**Table 1.** Social responsibility evaluation.

Left portion:

| | | | | |
|---|---|---|---|---|
| Employees' rights | Child labor | | | Use child labor or not |
| | | | | Child labor rate |
| | Labor compensation | Basic wage | | Minimum wage standard |
| | | | | Wage payment rate |
| | | | | Hour wage growth rate |
| | | | | Overtime wage payment rate |
| | | Additional salary preserve | | Labor insurance payment rate |
| | | | | Wage deduction ratio |
| | | Wage constitution | | Explicitly specify wage structure |
| | | | | Calculation method of overtime work payment |
| | Living conditions and working times | Labor safety production | | Production safety measures to construction |
| | | | Occupational disease | Disability series |
| | | | | Incidence rate |
| | | | Compensation | Safety accident casualty rate |
| | | | | Compensation situation |
| | | | Law performance | Labor law |
| | | | | Labor contract law |
| | | | | Employment romotion Law |
| | | | | Byelaw of inductrial injury insurance and other laws |
| | | Living and producing conditions | | Drinking water |
| | | | | Production site toilet conditions |
| | | | | Workplace air |
| | | | | Collective dormitory condition |
| | | Working-hours | | Working hours a week |
| | | | | Overtime hours a week |
| | | | | Most rest days a week |
| | | | | Off duty after 10p.m |
| | Dedication to employees | Contribution to employment | | New post rate |
| | | | | Human capital investment |
| | | | | Disabled post |
| | | | | Surplus labor force rate |

Right portion:

| | | | | | |
|---|---|---|---|---|---|
| Employees' rights | Dedication to employees | Satisfaction | | | Human capital maintenance capacity |
| | | | | | Profit margins per staff |
| | | | | | Labor productivity |
| | | Staff structure | | | Labor dispatch rate |
| | | | | | Family employees proportion |
| | | | | | Disabled employees proportion |
| | Human rights protection | Employee benefit | Employee rights | Freedom degree | Basic freedom — Voluntary overtime work |
| | | | | | Basic freedom — Resignation freedom |
| | | | | | Basic freedom — Nine-to-five free |
| | | | | | Refuse to arrest the gold and certificates |
| | | | | | Association and collective bargaining rights |
| | | | | | Organize trade unions and strike |
| | | | | Paid leave | paid leave |
| | | | | | days |
| | | | | Staff participation rate | Company management rate |
| | | | | | Union members participation rate |
| | | | Employee benefit | Training | Welfare payment rate |
| | | | | | Regular training |
| | | | | | Employee training rate |
| | | | | | Education funds per staff |
| | | | | | Union funds payment rate |
| | | Discrimination and punishment | Punitive measures | Punishment | Physical punishment |
| | | | | | Mental and physical stress |
| | | | | | Speech reproach |
| | | | | | Punishment to deduce wage |
| | | | | | Recessive abuse and unfair treatment |
| | | | Discrimination | | Religion discrimination |
| | | | | | Race discrimination |
| | | | | | Sex discrimination |
| | | | | | Age discrimination |

**Table 2.** Findings and limitations of different research methods.

| Method | Findings | Limitations |
|---|---|---|
| Delphi method | Strategies for improving social responsibility for companies in the food and pharmaceutical industries in Iran | Subjective and inefficient |
| Principal component analysis | The importance index of esg in Latin American listed companies is at a medium and high level | The weighting logic is not rigorous |
| Entropy value | Estimate the weight of each factor in the coal industry based on different parameters | Consider only statistical significance, but ignore practical significance |
| Analytic hierarchy process | Calculate the scores of different companies in the coal industry | Subjective and computationally expensive |

3.2.2. Calculate the Mutation Fuzzy Membership Function

The most common catastrophe evaluation indicator system of a mutation system has three types, which are sharp point mutation system, dovetail mutation system and butterfly mutation system. The sharp point mutation system can be divided into two child indexes. The model of a sharp point mutation system is:

$$f(x) = x^4 + ax^2 + bx \tag{1}$$

The model of a dovetail mutation system is:

$$f(x) = \frac{1}{5}x^5 + \frac{1}{3}ax^3 + \frac{1}{2}bx^2 + cx \tag{2}$$

The model of a butterfly mutation system is:

$$f(x) = \frac{1}{6}x^6 + \frac{1}{4}ax^4 + \frac{1}{3}bx^3 + \frac{1}{2} + cx^2 + dx \tag{3}$$

$f(x)$ represents the potential function of the system state variables $x$, the state variables $a, b, c$ and $d$ are the controlled variable of the state variables.

The model's equations based on catastrophe system could deduce the normalization equation of each catastrophe system. According to the catastrophe theory, all critical points of $f(x)$ in the catastrophe system make an equilibrium surface; all critical points can be obtained through the first derivative of $f(x)$, that is:

$$f'(x) = 0 \tag{4}$$

Its singular point can be obtained through the second-order derivative to $f(x)$, that is:

$$f''(x) = 0 \tag{5}$$

The equation of divergent sets shows that when the control variables satisfy the equation, mutations occur in the evaluation index system, then the use of the multidimensional fuzzy membership function principle normalizes the formula. Take the cusp catastrophe model as an example:

$$f'(x) = 4x^3 + 2ax + b = 0 \tag{6}$$

$$f''(x) = 6x^2 + a = 0 \tag{7}$$

So, finishing the above two equations can obtain the divergent set of the cusp catastrophe model:

$$a = -6x^2, \quad b = 8x^3 \tag{8}$$

Similarly, the divergent set of the swallowtail catastrophe model and butterfly catastrophe model are as follows:

The divergent set of the swallowtail catastrophe model:

$$a = -6x^2, \quad b = 8x^3, \quad c = -3x^4 \tag{9}$$

The divergent set of the butterfly catastrophe model:

$$a = -10x^2, \quad b = 20x^3, \quad c = -15x^4 \tag{10}$$

Above that, the potential function of the system state variables $x$, the state variables in front of $a$, $b$, $c$ and $d$ are the controlled variable of the state variables. In order to facilitate the evaluation of the target, the bifurcation set equation is normalized to get the catastrophe fuzzy membership function. In order to take advantage of the catastrophe progression method, the values of state variables and control variables should be unified, therefore the bifurcation set of the catastrophe model can be written as the following functions:

After deformation, the bifurcation set of the cusp catastrophe model is:

$$x_a = \sqrt{\frac{a}{-6}}, \quad x_b = \sqrt[3]{\frac{b}{8}} \tag{11}$$

After deformation, the divergent set of the swallowtail catastrophe model is:

$$x_a = \sqrt{\frac{a}{-6}}, \quad x_b = \sqrt[3]{\frac{b}{8}}, \quad x_c = \sqrt[4]{\frac{b}{-3}} \tag{12}$$

After deformation, the divergent set of the butterfly catastrophe model is:

$$x_a = \sqrt{\frac{a}{-10}}, \quad x_b = \sqrt[3]{\frac{b}{20}}, \quad x_c = \sqrt[4]{\frac{b}{-15}} \tag{13}$$

In the formula, $x_a$, $x_b$, $x_c$ and $x_d$ are views corresponding to the $x$ value of $a$, $b$, $c$ and $d$.

If $|x| = 1$, we obtain $|a| = 6$ and $|b| = 8$ in the cusp catastrophe model; $|a| = 6$, $|b| = 8$ and $|c| = 3$ in swallowtail catastrophe model; $|a| = 10$, $|b| = 20$, $|c| = 15$ and $|d| = 4$ in the butterfly catastrophe model. This will determine the value range of the absolute value of state variables and the controlled variable. The absolute values for value range are respectively: if $0 \leq |x| \leq 1$, we obtain $0 \leq |a| \leq 6$ and $0 \leq |b| \leq 8$ in the cusp catastrophe model; $0 \leq |a| \leq 6$, $0 \leq |b| \leq 8$ and $0 \leq |c| \leq 3$ in swallowtail catastrophe model; $0 \leq |a| \leq 10$, $0 \leq |b| \leq 20$, $0 \leq |c| \leq 15$ and $0 \leq |d| \leq 4$ in the butterfly catastrophe model. However, these values $x$, $a$, $b$, $c$ and $d$ cannot be unified. For the actual computing method, and reducing the relative range, this does not affect the nature of the catastrophe model, the absolute value of the state and control variable values simplify to 0–1. So, $a$, $b$, $c$ and $d$ divided by the maximum value can reduce $a$, $b$, $c$ and $d$ to 0–1.

The normalized formula of the catastrophe system model can be obtained as follows:

Normalized formula of cusp catastrophe:

$$x_a = a^{\frac{1}{2}}, \ x_b = b^{\frac{1}{3}} \tag{14}$$

Normalized formula of swallowtail catastrophe:

$$x_a = a^{\frac{1}{2}}, \ x_b = b^{\frac{1}{3}}, \ x_c = c^{\frac{1}{4}} \tag{15}$$

Normalized formula of butterfly catastrophe:

$$x_a = a^{\frac{1}{2}}, \ x_b = b^{\frac{1}{3}}, \ x_c = c^{\frac{1}{4}}, \ x_d = d^{\frac{1}{5}} \tag{16}$$

After the normalization processing, the values of the state variables and control variables are 0–1. These mutations are called the fuzzy membership function, the core mutations' multi-criteria evaluation method.

### 3.2.3. Determine the Function to Obtain the Sorted Evaluation Results

The evaluation of social responsibility is a multi-level multi-index rating system, there are positive indicators, reverse indicators and moderate indicators. In the evaluation process, the indicators are trended at first, which means all kinds of indexes translate into positive indicators. Reverse indicators translate into positive with the following formula:

$$x'_{ij} = \max(x_{ij}) - x_{ij} \ (1 \leq i \leq n) \tag{17}$$

Moderate indicators translate into positive with the following formula, where $l$ means the moderate value.

$$x'_{ij} = \max|x_{ij} - l| - |x_{ij} - l| \ (1 \leq i \leq n) \tag{18}$$

Secondly, indicators are trended in order to meet the requirements of normalized data between 0–1 in the catastrophe progression method, then the positive indicators are standardized to be dimensionless. The uniform trend of indicators can be standardized by the following formula:

$$p_{ij} = \frac{x_j - \min x_j}{\max x_j - \min x_j} \tag{19}$$

According to the multidimensional fuzzy membership function's principle, we use each average sub-index to obtain the evaluation objectives under the control variable in the same target. After the standardization of the evaluation of the social responsibility, we can take advantage of the mutation system model to evaluate the corporate social responsibility to employees. The scores from large to small correspond to the ranking of corporate social responsibility performance from high to low.

## 4. Results

The Chinese clothing industry is a very important traditional manufacturing industry with labor-intensive, low-skilled and low-capital characteristics. In recent years, it has been growing at a high speed in China and its industrial structure and its product structure have been optimized step by step and its industrial gradient and technical level have been improving continuously. Therefore, Chinese textile trade status in the world is rising and China has become the world's first producer and exporter of textiles and clothing. In recent years, China's foreign trade clothing has been often greatly influenced by the Western green standard (ISO14000) and ecological textile standard (Oeko-Tex standard 100) block. Since 2007, the China National Textile and Apparel Council (CNTAC) has cooperated with the International Labor Organization (ILO) and United Nations Industrial Development Organization (UNIDO) to implement the "China textile enterprise social responsibility project" and take on trials in some enterprises, effectively promoting the social responsibility management system CSC9000T, and achieved positive results. This showed that the level of social responsibility the Chinese clothing industry has is high, being well representative of the Chinese manufacturing industry. Therefore, this article illustrated the principles and methods of employee social responsibility evaluation by choosing the top 100 clothing enterprises in China in 2018 as a sample. Let us take Youngor Group as an example, the calculation steps are as follows:

Firstly, the reverse index and moderate index need trend change. This article has 16 reverse indexes, four appropriate indicators, including six reverse son indexes and three moderate son indexes beneath the rights and interests of employees, ten reverse son indexes and a moderate son index beneath the guarantee of human rights. We use Formulas (17) and (18) to trend the reverse son index and moderate son index.

For the incidence of occupational diseases, for example, we make reverse indicators positive: the maximum value of the incidence of occupational disease index is Bo Si Deng Co., Ltd., about 0.46%; the incidence of occupational diseases of Youngor Group is about 0.46%.

According to Formula (20), the incidence of occupational disease index after the positive value of the Youngor Group is 0.06%. Take the example of weekly work hours, for example, we make the moderate index positive: the value of weekly work hours is 48 h, meaning a maximum of $|x_{ij} - 1| = 18$, which is Qingdao JiFa Group Co., Ltd., the weekly work hours of the Youngor Group is 62 h, meaning $|x_{ij} - 1| = 14$, so the value of Youngor Group's weekly work hours after processing is 4.

Then, all the indexes need a trend change using Formula (19). Take incidence of occupational diseases as an example, the minimum value of incidence of occupational disease is BoSiDeng Group Co., Ltd., about 0, the maximum value of incidence of occupational disease is ChangZhou LiHuaDa Group Co., Ltd., about 0.35%, so the standard value of incidence of occupational disease of the Youngor Group is:

$$\text{Standard value} = \frac{0.0006 - 0}{0.0035 - 0} = 0.1714 \tag{20}$$

Finally, we need to use the method of the mutation series of the sharp point mutation system model, dovetail mutation system model and the butterfly mutation system model of the normalization formula to operate the standardized data. Because of all indicators becoming positively changed indexes, we need the numerical averaging of $x_a$, $x_b$, $x_c$ and $x_d$ using the formula:

$$x = \frac{\{x_a, \ x_b, \ x_c, \ x_d\}}{4} \tag{21}$$

With incidence of occupational diseases, for example, occupational disease can be divided into disability progression after occupational disease and the incidence of occupational diseases, so the occupational disease can be applied by the cusp catastrophic model which uses the normalization formula:

$$x_a = a^{\frac{1}{2}}, \ \ x_b = b^{\frac{1}{3}}, \ \ x_a = 0.85^{\frac{1}{2}} = 0.9220, \ \ x_b = 0.1714^{\frac{1}{3}} = 0.5555 \tag{22}$$

According to the formula:

$$x = \sum \{x_a, \ x_b\}/2 = \sum \{0.09220, 0.5555\} = 0.7388 \tag{23}$$

This means the value of the incidence of occupational diseases of the Youngor Group is 0.7338. Next, we calculate each index of the Youngor Group and finally conclude the value of the employees' rights indicator and the value of the human rights protection indicator, which are 0.6001 and 0.3447. Then, we use the cusp catastrophic model, which is applied with the normalization Formula (14), to normalize the employees' rights indicator and the human rights protection indicator, which values are 0.7963 and 0.9134, so the final score of social responsibility to employees of the Youngor Group is:

$$x = \sum \{x_a, \ x_b\}/2 = \sum \{0.07963, 0.9134\} = 0.8549 \tag{24}$$

Other enterprises' social evaluation process can be calculated similarly, and the results are shown in Table 3.

It can be seen from the results that the top 100 clothing companies in China all maintain a high level of above 0.69, and the CSR Rank is generally close to the Industry Rank.

**Table 3.** CSR ranking of the top 100 companies in the industry.

| CSR Rank | Industry Rank | Corporate Name | Score | CSR Rank | Industry Rank | Corporate Name | Score | CSR Rank | Industry Rank | Corporate Name | Score | CSR Rank | Industry Rank | Corporate Name | Score |
|---|---|---|---|---|---|---|---|---|---|---|---|---|---|---|---|
| 1 | 6 | Shanghai Kai Kai | 0.8707 | 26 | 21 | Jiangsu wizhong industrial Co., Ltd. | 0.8278 | 51 | 76 | Shijiazhuang 3502 factory | 0.8798 | 76 | 68 | Zhejiang Kaier Clothes | 0.7586 |
| 2 | 18 | Baoxiniao Group | 0.8706 | 27 | 26 | Jiangsu Yalu Group | 0.8260 | 52 | 77 | Zhejiang Langwei Group | 0.8765 | 77 | 69 | Zhejiangeyu Group | 0.7575 |
| 3 | 2 | Hongdou Group | 0.8705 | 28 | 44 | Bu Sen Group | 0.8177 | 53 | 78 | Shanxi Weiye Co., Ltd. | 0.8734 | 78 | 94 | Chongqin 3533 Clothes | 0.7561 |
| 4 | 10 | Roman Group | 0.8685 | 29 | 35 | Zhejiang Shouwang Group | 0.8111 | 54 | 25 | Peacebird Group | 0.8712 | 79 | 96 | Zhejiang babei tie Co., Ltd. | 0.7542 |
| 5 | 3 | Heilan Group | 0.8684 | 30 | 42 | Lanyan Group | 0.8001 | 55 | 61 | Yaya Group | 0.8702 | 80 | 89 | Yuandong industrial | 0.7531 |
| 6 | 13 | Rouse Group | 0.8674 | 31 | 51 | Hangzhou Dali industrial Co., Ltd. | 0.7890 | 56 | 35 | Yefeng Group | 0.8701 | 81 | 91 | Anhui Hongrun Group | 0.7499 |
| 7 | 8 | Boston | 0.8660 | 32 | 60 | Changzhou Huali Group | 0.7887 | 57 | 28 | Shandong Daiyin Co., Ltd. | 0.8678 | 82 | 75 | Jiangsu Leino Group | 0.7462 |
| 8 | 4 | Shan Shan | 0.8643 | 33 | 54 | Zhejiang Huating Co., Ltd. | 0.7867 | 58 | 81 | Changzhou Jinsong Group | 0.8513 | 83 | 98 | Wuhan Hongren Group | 0.7421 |
| 9 | 14 | Hemboug Group | 0.8628 | 34 | 31 | Wensli Group | 0.7851 | 59 | 30 | Aiyinei Group | 0.8451 | 84 | 85 | Zhejiang Qingsheng Clothes | 0.7381 |
| 10 | 7 | Ji Fa Group | 0.8622 | 35 | 22 | Dapai Group | 0.7847 | 60 | 32 | Ningbo Progen Co., Ltd. | 0.8236 | 85 | 86 | Zhejiang Dali Group | 0.7349 |
| 11 | 15 | Dayang Group | 0.8621 | 36 | 70 | Zhejiang Juyin Co., Ltd. | 0.7821 | 61 | 33 | Yekiya Group | 0.8104 | 86 | 92 | Quanzhou Green Group | 0.7338 |
| 12 | 11 | Metersbonwe Group | 0.8617 | 37 | 47 | Dadi Group | 0.8170 | 62 | 37 | Shandong Sunshell Group | 0.7923 | 87 | 71 | Weihai Huayu & Beijing Jingbei | 0.7334 |
| 13 | 19 | Shanghai three gun Group | 0.8556 | 38 | 49 | Hanbo (China) Group | 0.8099 | 63 | 41 | Beijing Xuelian Group | 0.7851 | 88 | 73 | Jiangsu Liutan Group | 0.7324 |
| 14 | 1 | Youngor group | 0.8549 | 39 | 65 | Zhejiang Xinchengda investment | 0.8034 | 64 | 43 | Semir Group | 0.7754 | 89 | 82 | Yancheng Yuren Group | 0.7319 |
| 15 | 17 | Shandong Uniform Co., Ltd. | 0.8528 | 40 | 58 | Jiangsu Henwei industrial Co., Ltd. | 0.8011 | 65 | 46 | Zhejiang Kobron Group | 0.7746 | 90 | 83 | Guangdong DKD Group | 0.7310 |

**Table 3.** *Cont.*

| CSR Rank | Industry Rank | Corporate Name | Score | CSR Rank | Industry Rank | Corporate Name | Score | CSR Rank | Industry Rank | Corporate Name | Score | CSR Rank | Industry Rank | Corporate Name | Score |
|---|---|---|---|---|---|---|---|---|---|---|---|---|---|---|---|
| 16 | 12 | Jiangsu Hubao Group | 0.8524 | 41 | 72 | Shandong Aoshi Group | 0.8004 | 66 | 48 | Jiangsu Dongdu Group | 0.7709 | 91 | 95 | Shandong Shengge Group | 0.7239 |
| 17 | 9 | Zhejiang Fukeda Group | 0.8509 | 42 | 67 | Hudu Group | 0.8977 | 67 | 51 | Huizhou Fusen industrial | 0.7678 | 92 | 99 | Jianjingfeng group | 0.7235 |
| 18 | 5 | Matsuoka Group | 0.8488 | 43 | 63 | Huashi (China) Group | 0.8901 | 68 | 52 | Zhejiang Shenying Group | 0.7671 | 93 | 74 | Qingnao Hongling Group | 0.7233 |
| 19 | 20 | Xinlang Co., Ltd. | 0.8464 | 44 | 80 | Shenzhen Huasi Co., Ltd. | 0.8867 | 69 | 53 | Qingdao Haishan | 0.7668 | 94 | 100 | Yunan Aodiluo industrial | 0.7227 |
| 20 | 16 | Chen Feng group Co., Ltd. | 0.8404 | 45 | 88 | Bailide Co., Ltd. | 0.8832 | 70 | 55 | Beijing Tongniu Group | 0.7658 | 95 | 97 | Zhejiang Haipo Group | 0.7222 |
| 21 | 29 | Handan Xuechi Group | 0.8398 | 46 | 45 | Fujian Tries Group | 0.8826 | 71 | 57 | Fuguiniao Group | 0.7651 | 96 | 93 | Shanghai Chunzu | 0.7217 |
| 22 | 23 | ZhuangJi Group | 0.8366 | 47 | 24 | Weixing Group | 0.8821 | 72 | 59 | Jiangsu Sanyou Group Co., Ltd. | 0.7641 | 97 | 90 | Huning Jingcai Clothes | 0.7215 |
| 23 | 34 | Shandong Xianxia Group | 0.8356 | 48 | 56 | Guangdong Leiyi Co., Ltd. | 0.8818 | 73 | 62 | Quanzhou Longquan Clothes | 0.7634 | 98 | 87 | Zhejiang Taizilong Group | 0.7191 |
| 24 | 38 | Shanghai Kaituo Co., Ltd. | 0.8298 | 49 | 27 | AB Group Co., Ltd. | 0.8805 | 74 | 64 | Zhejiang Jinsanfa | 0.7621 | 99 | 84 | Shanghai Hailuo Group | 0.6991 |
| 25 | 41 | K-boxing Group | 0.8282 | 50 | 39 | Wuhan Aidi Group | 0.8802 | 75 | 66 | Wuxi Guangning Group | 0.7617 | 100 | 79 | Hunan Xintai Group | 0.6911 |

## 5. Discussion

Judging from the social responsibility situation of employees of the top 100 enterprises in China's clothing industry, the scores of each clothing company are relatively high, indicating that China's clothing companies have performed their social responsibilities well to employees. Among the companies we evaluated, the top 20 companies were still in the top 20 in terms of social responsibility for their employees, with a score of over 0.846.

After a preliminary look at the financial health of these 100 companies, we found that the top 20 companies generally have good financial performance and relatively stable cash flow positions. According to the research of Akbar et al., companies in the mature stage have more stable cash flow and correspondingly lower bankruptcy risk [43]. Conversely, it can be preliminarily judged from the stable cash flow performance of the above 20 companies that they have entered the maturity stage of the life cycle.

In addition, according to common sense, the company's life cycle has entered a mature stage, which means that these companies have more time, energy and financial resources to improve employee treatment and fulfill social responsibilities [44], thereby improving the company's reputation, influence and attractiveness. The research of Sun et al. also pointed out that CSR will inhibit financing constraints in the mature stage of enterprises [45] and then affect the financial situation.

Correspondingly, companies with low CSR rankings tend to be in the growth stage of their life cycle. For them, surviving in the fierce market competition is the most important issue for enterprises to consider. They often need to devote more resources to occupy a larger market share, which also leads to unstable or even negative cash flow, so they do not have the time or ability to care about employees' fulfillment of corporate social responsibility. As expressed by another conclusion of Sun et al.: for enterprises in the growth stage or early stage of development, excessive fulfillment of CSR will bring them more financial constraints [45].

Through the above analysis, it can be found that the financial performance and CSR performance of enterprises are interacting to a certain extent, and the specific roles will be different due to the different life cycles of enterprises. For companies in the start-up or growth stage, corporate social responsibility performance and financial performance will contain each other, while for companies in the mature and declining stages, there is a positive correlation between them.

## 6. Conclusions

This article establishes an evaluation system of corporate social responsibility to employees in the Chinese clothing industry through the catastrophe progression method and evaluates and ranks the top 100 Chinese fabric companies in 2018.

For labor-intensive enterprises, the ranking results of this study show that the overall ranking is similar to the CSR ranking, which preliminarily proves that there is a positive relationship between CSR and the development stage of the enterprise of the clothing industry.

The research result also shows that the CSR scores of the top 100 clothing enterprises in China are all between 0.69 and 0.88. It can be preliminarily judged that although Chinese enterprises are not greatly influenced by labor unions, and most of them rely on corporate consciousness to fulfill their social responsibilities to employees, as far as the top 100 clothing enterprises are concerned, their CSR performance is at the middle and upper levels. However, due to the lack of evaluation and comparison of foreign enterprise samples, this conclusion needs to be further confirmed by follow-up research.

The catastrophe progression method adopted in this study considers the different influences of various subjective and objective factors on corporate social responsibility, and assigns different weights to the 61 subdivision elements, making the results more realistic.

We believe that it is beneficial and necessary to establish an enterprise's employee social responsibility evaluation system, and the accuracy of this system has also been preliminarily confirmed in this case.

However, the research also has limitations, such as the inability to rigorously prove the accuracy of the evaluation system, not to point out the degree of influence of different factors, and the establishment of the system has little guiding significance for enterprise managers.

In the follow-up research, it is recommended to further improve the enterprise's social responsibility evaluation system for employees on this basis, add other suitable quantitative indicators, and modify the factors that are not suitable or repeated. In addition, the accuracy and feasibility of the model can be further confirmed by combining it with the employee satisfaction survey. The specific situation of different industries can also be compared and analyzed, and the commonalities and characteristics between industries can be extracted to improve the comparability of CSR scores in different industries.

**Author Contributions:** Conceptualization, Y.J.; methodology, W.Z.; validation, Y.J. and Y.T.; formal analysis, W.Z.; investigation, Y.T.; resources, Y.Z.; data curation, Y.Z.; writing—original draft preparation, Y.J.; writing—review and editing, W.Z. and Y.T. All authors have read and agreed to the published version of the manuscript.

**Funding:** This work was supported by the support of China National Social Science Fund Projects (No. 15BGL054), the Jiangsu Province Social Science Application Research Excellent Engineering Project (No. 21SYC-028); the Research Project of Philosophy and Social Science in Jiangsu Universities (No. 2019SJA0476).

**Institutional Review Board Statement:** Not applicable.

**Informed Consent Statement:** Not applicable.

**Data Availability Statement:** Not applicable.

**Acknowledgments:** We would like to thank the support of the Guidance Project of Jiangsu Philosophy and Social Science Research Foundation for University.

**Conflicts of Interest:** The authors declare no conflict of interest.

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
