# Peer review of "The Construction of an Evaluation System of Corporate Social Responsibility to Employees: An Empirical Study in the Chinese Clothing Industry"

_sustainability, doi:10.3390/su141610215_

Round 1

Reviewer 1 Report

Thank you for your interesting work on 'The Construction of Evaluation System of Corporate Social Responsibility to employees: an Empirical Study in Chinese ' Clothing Industry'. The context is very interesting and topical.

It would be useful to provide at least 3 clear contributions of the paper to the current literature.

In the discussion part, I am not clear as to how you have decided that some companies are in the mature/growth stage of the life cycle. However, you suggest stable cash flow for the former, but to me, it is not very clear how this can be replicated. Also you need to expand on your discussion.

Need to pay considerable attention to the referencing using the style of MDPI - sustainability journal both within the text and the final reference list and ensure that all references in text are also shown in the final reference list, for example, Dodd and Berle in line 127.

It would be interesting to develop a table with the different research methods and their findings and limitations to be included to justify the different methods in lines 227-228.

I am not clear as to how the data is collected. Can the authors make this clear.

Can you ensure that any abbreviations are consistent, e.g. csr Line 111.

The paper can be improved by checking the grammar, e.g. "the Dodd and Berle..." Line 127

Author Response

Dear reviewer,

    First of all, thank you very much for your time and hard work for this review. Secondly, you were very professional and serious in the review process, pointed out a series of questions in the paper, and provided great help for us to revise and improve our article. Lastly, Finally, let us thank you again for your insightful thoughts and enlightening questions. Please find the attachment for our response to your comments.

   Best Regards

   Yongjun Tang

Reviewer 2 Report

Thank you for having the chance to review this paper that has promise in regards to analyse quantitative analysis methods as a means for advancing research and practice in the field. However, this topic is not sufficiently tackled in the manuscript. Moreover, and this is also my main point of critique, the logical line of argumentation is very weak and not adequate for the reader 

The paper has promise in regards to its attempt to analyse quantitative analysis methods as a means for advancing research and practice on the field. However, and this is my main point of critique, the logical line of argumentation is very weak and not adequate for the reader to understand why and how can help in this. There are no clear references to the literature, in particular when introducing fundamental concepts for the research 

 The methodology is appropriate 

The paper needs major work to convince that it makes a worthy contribution 

 I would suggest the author(s) to have the text reviewed by a native English speaker. 

I suggest enriching the bibliography with the following reference:
VALERI M. (2021), Organizational Studies. Implications for the Strategic Management, Springer, Switzerland 

Author Response

Dear reviewer,

Thank you for your constructive comments and questions. Your comments and questions have brought great help to the revision and improvement of our article. Please allow us to respond to your comments one by one and refer to the attachment for details.

Best Regards

Yongjun Tang

Reviewer 3 Report

The article addresses the important issue of corporate social responsibility towards employees. This is of particular relevance to the clothing industry, which is the subject of the research undertaken by the authors.

Abstract:

In the abstract, the authors outline the background to the article and the significance of the issue undertaken. Highlighted is the authors' use of the catastrophe progression method to assess corporate social responsibility towards the employee. The section also draws the main conclusions from the research. However, the authors have not clearly defined what is the purpose of the article, of the research undertaken. It is therefore suggested that the aim be made more specific.

1.         Introduction:

The introduction provides an overview of the scope of the study and the main issues addressed in the paper. However, it is suggested that the information on the research subject and the research methods used should be made more precise.

2. Literature Reviews and 3. Theoretical Basis

In my opinion, the description of the theoretical framework was unnecessarily separated into two chapters: 2. Literature Reviews and 3. Theoretical Basis. I propose to consolidate the two chapters into one representing the presentation of the theoretical framework. I do not understand the advisability of this division.

In this part, the authors, based on the literature analysis, refer to the discussion and explanation of the main issues raised in the article. What is missing in this section, in my opinion, however, is a justification based on the literature analysis for the research undertaken and the identification of
a gap as well as an emphasis on originality, novelty. The references in my opinion need to be deepened. The authors rely on current literature but only on 35 literature items.

4. Method

This section of the paper lacks basic information on the methodology, the methods, techniques and tools used and a description of the research process. There is also a lack of information on the research subject, its characteristics and the research sample.

The research subject is only mentioned briefly and only in section 5 Results. It is important to complete all this information.

Table 1 is unclear and lacks a summary comment under the table.

Line 187 - the word "Modic" ?

In many places the necessary spaces are missing among others: line 205-212 lacks the spacing done.

5. Results.

This is a valuable and interesting part of the work presenting the results of the research carried out.

However, I have the following comments:

- An explanation, a summary, is missing under Table 2.

- Lines 333-334 are missing spaces.

6. Discussion

The section presents interesting conclusions. In my opinion, however, this section lacks an in-depth discussion. This may be due to the previously indicated underdetermination of the purpose of the work, the novelty, the methodology and the subject of the research.

7. Conclusions

The authors have synthesised the conclusions in this section and have also indicated the limitations of the research. It is suggested to add information on the wider applicability of the research and to indicate future research directions.

Author Response

Dear reviewer,

Thank you for your constructive and valuable comments. Your comments and questions have brought great help and inspiring to the revision and improvement of our article. Please allow us to respond to your comments point by point and refer to the attachment for details.

Round 2

Reviewer 1 Report

The authors need to make the script more readable for publication:

Lines 51 and 52 - there are quotations but no citations. Please insert citations. I suggest that you go through the paper and ensure no further citations are missing.

Lines 143 to 146 - the sentence needs to be rewritten as it is in poor English and has grammar errors.

Line 163 please insert the following: disclosure of the information

Line 185: you show"28" - is this an additional citation?

I assume that the Language Editing Services will sort these issues for the language and grammar errors. Lines 188, 212, 213, 229, 242, 430 etc.

Conclusion: please refer to the original research question/aim and provide a paragraph of how this has been answered.

Author Response

Dear Reviewer,

     Thank you for your kindness and care. The questions you raised are of great help to us in revising and improving the manuscript. We do our best to eliminate errors and improprieties in order to improve the quality of the manuscript. In order to better explain our modification process to you, please check the attachment.

    Thank you again for your good comments!

Best Regards

Yongjun Tang  et al.
